# The impact of poly-A microsatellite heterologies in meiotic recombination

Angelika Heissl[1], Andrea J Betancourt[2], Philipp Hermann[3] , Gundula Povysil[4], Barbara Arbeithuber[1] , Andreas Futschik[3], Thomas Ebner[5], Irene Tiemann-Boege[1]

Meiotic recombination has strong, but poorly understood effects on short tandem repeat (STR) instability. Here, we screened thousands of single recombinant products with sperm typing to characterize the role of polymorphic poly-A repeats at a human recombination hotspot in terms of hotspot activity and STR evolution. We show that the length asymmetry between heterozygous poly-A's strongly influences the recombination outcome: a heterology of 10 A's (9A/19A) reduces the number of crossovers and elevates the frequency of non-crossovers, complex recombination products, and long conversion tracts. Moreover, the length of the heterology also influences the STR transmission during meiotic repair with a strong and significant insertion bias for the short heterology (6A/7A) and a deletion bias for the long heterology (9A/19A). In spite of this opposing insertion-/deletion-biased gene conversion, we find that poly-A's are enriched at human recombination hotspots that could have important consequences in hotspot activation.

## Introduction

Repeated stretches of DNA with 1–6 nt per repeat unit and varying tract lengths, known as short tandem repeats (STRs) or microsatellites, comprise about 3% of the human genome (Lander et al, 2001). STRs have attracted attention because of their association with diseases (e.g., cancer and neurological disorders) (reviewed in (Pearson et al, 2005; Richard et al, 2008; Boland and Goel, 2010; Shokal and Sharma, 2012; Polyzos and Mcmurray, 2017)). STR polymorphisms also play a role in gene expression variation (Gymrek et al, 2016), alternative splicing (Hui et al, 2005), chromatin packaging (Kirkpatrick et al, 1999), and nuclear organization (Pathak et al, 2013) (reviewed also in Bagshaw (2017)). In spite of the functional importance of STRs, our understanding of the drivers of their evolution is far from complete (reviewed in Ellegren (2004), Richard et al (2008)), with mononucleotide repeats, the second most common repeat type in the human genome, particularly neglected because of the difficulty of accurately determining their lengths (reviewed in Ellegren (2004), Zavodna et al (2018)).

We know that STRs are highly mutable, with a reported mutation rate of $10^{-4}$ to $10^{-3}$ insertion/deletions (indels) per nucleotide per generation, partly dependent on repeat length, motif type, and repeat purity (Sun et al, 2012; Fungtammasan et al, 2015). The widely accepted mechanism proposed to explain this instability invokes a slippage mechanism, in which STRs gain or lose repeats because of strand slippage during replication (Sia et al, 1997). However, STRs are not only unstable during replication but also during meiotic and mitotic homologous recombination and post-replication DNA repair, implying additional mechanisms beyond the simple slippage model (reviewed in Richard et al (2008), Bagshaw (2017)).

Although potentially an important factor in their evolution, STR instability in the context of meiotic recombination is still far from understood, and the existing evidence is patchy and conflicting among studies. Naively, if recombination is mutagenic for STRs, we would expect a positive correlation between STR variation and recombination. Early work found that broad-scale recombination rates, however, were most weakly correlated (Payseur & Nachman, 2000; Kong et al, 2002; Jensen-Seaman et al, 2004), although a more recent study found a slight positive correlation, with the strongest effect observed for mono- and dinucleotides (Mallick et al, 2016). A study using fine-scale maps of recombination hotspots suggested that recombination itself may only weakly drive this correlation, instead implicating local base composition (Brandstrom et al, 2008).

Nevertheless, the link between STRs and recombination merits further study because several lines of evidence suggest that meiosis affects repeat instability. In particular, meiotic recombination is the most frequent destabilizing process for minisatellite repeats (unit size ~15–100 bp) (Jeffreys et al, 1998; Bois & Jeffreys, 1999; Bishop & Schiestl, 2000). Moreover, meiotic recombination and some STRs, especially

---

[1]Institute of Biophysics, Johannes Kepler University, Linz, Austria   [2]Insitute of Integrative Biology, University of Liverpool, Liverpool, UK   [3]Institute of Applied Statistics, Johannes Kepler University, Linz, Austria   [4]Institute of Bioinformatics, Johannes Kepler University, Linz, Austria   [5]Department of Gynecology, Obstetrics and Gynecological Endocrinology, Kepler University Clinic, Linz, Austria

Correspondence: Irene.tiemann@jku.at
Gundula Povysil's present address is Institute for Genomic Medicine, Columbia University, New York, USA
Barbara Arbeithuber's present address is Department of Biology, Pennsylvania State University, University Park, USA

poly-A's, may be functionally related. Certain STRs or STR-containing motifs, including poly-A's, are overrepresented in humans (Myers et al, 2005), yeast (Bagshaw et al, 2008), and *Arabadopsis* (Horton et al, 2012; Choi et al, 2013, 2018; Wijnker et al, 2013) recombination hotspots (reviewed in Tiemann-Boege et al, 2017). The interpretation of these findings, however, is confounded by the overlap between recombination hotspots and transcription sites in the latter two species. Nevertheless, direct experimental evidence suggests that repeats can promote recombination activity: deletion of a poly-A in one hotspot in yeast reduced meiotic conversion activity (Schultes & Szostak, 1991), whereas an introduced GT dinucleotide repeat increased recombination (Treco & Arnheim, 1986). The presence of repeats can also interfere with recombination, as seen for an introduced long GT repeats which inhibited crossover (CO) formation, and the repeat itself was highly unstable (Gendrel et al, 2000).

In addition, meiotic gene conversion has been implicated as an important driver of STR evolution, in particular, for disease-causing trinucleotide microsatellite (Jankowski et al, 2000). In fact, meiotic gene conversion may affect all types of indels, not just STRs. Gene conversion biases appear to favor the long over the short allele in small indels (insertion-biased gene conversion; iBGC) (Ometto et al, 2005; Presgraves, 2006; Leushkin & Bazykin, 2013). In contrast, the analysis of COs and non-crossovers (NCOs) of a very large human pedigree showed that the shorter allele was transmitted more often than the longer one in gene conversions (Halldorsson et al, 2016). A bias towards the shorter allele was also reported for nonallelic gene conversions in *Drosophila* (Assis & Kondrashov, 2012). Because none of these studies differentiate between the type and length nor the nature of the indel (e.g., repeat type), it is difficult to compare and interpret the different trends between studies.

STR instability in the context of meiotic recombination has been underexplored, and thus, models are incomplete, especially for mononucleotide repeats. Here, we screened thousands of single CO and NCO products from a human recombination hotspot using pooled sperm typing and characterized the transmission of two polymorphic poly-A repeats. We explored the effect of these STRs on CO and NCO rates and on biases of allelic transmission by gene conversion. Our unique, high-resolution data set provided important insights in the effect of length asymmetry and heterozygosity of STRs in double-strand break (DSB) repair and aided to detangle the complex relationship between STR instability and meiotic recombination.

## Results

### Hotspot features dissected with pooled sperm typing

We analyzed the sequence of single meiotic products from a human recombination hotspot located on chromosome 16 within the RBFox 1 intron (hotspot HSII) with an average SNP density of 4 SNPs/kb. There are three poly-A repeats in the vicinity (6A/7A, 9A/19A, and 23A) located −490 bp, −154 bp, and +239 bp from the first base of a central PRDM9 motif at chromosome position chr16: 6,361,057–6,361,088 (GRCh37/hg19) shown to actively bind PRDM9 (Altemose et al, 2017) (Fig 1A). Single meiotic products were collected using pooled sperm typing as previously described

(Tiemann-Boege et al, 2006; Arbeithuber et al, 2015). This method amplifies single CO or NCO molecules in aliquots containing low concentrations of sperm DNA by allele-specific, nested quantitative PCR. CO and NCO products were collected based on similar principles, with the main difference in the initial primer setup (Fig 1B). Briefly, to retrieve COs, we used nested PCR with two flanking pairs of allele-specific PCR primers targeting four informative (heterozygous) SNPs, two on each side of the hotspot. This design preferentially amplified single CO products that were then characterized by genotyping heterozygous internal SNPs with allele-specific PCR or TaqMan. To collect NCOs, we selectively amplified only one of the two homologues using external flanking SNPs, followed by a second allele-specific quantitative PCR targeting internal informative SNPs. To normalize for differences in the DNA quality, inherent assay conditions, and experimental variation, we screened in each experiment the number of "amplifiable sperm" and estimated CO and NCO frequencies considering this correction factor (see the Materials and Methods [Testing the number of amplifiable genomes] section of the Supplementary Information).

We recovered eight informative donors that were either heterozygous (Ht; 9A/19A) or homozygous (Ho; 19A/19A) for one of the central poly-A's (Table S1). We were not able to collect recombinant molecules from 6A/6A or 9A/9A Ho donors, because this genotype was linked only to homozygous flanking markers that were inadequate for our pooled sperm typing assay. We verified that our donors, all of European descent, are carriers of the most common West Eurasian allele of PRDM9 (variant A), a *trans*-acting factor that determines the placement of DSBs at the onset of meiosis. COs and NCOs for both reciprocal products were characterized in these eight different donors, and we collected in total 4,448 COs from 3,948,418 amplifiable sperm and 246 NCOs from 360,474 amplifiable sperm (Tables S2–S4).

We estimated the location of the hotspot center from our CO and NCO data as the region with the highest concentration of breakpoints, which lies within a hotspot estimated from linkage disequilibrium data (International Hapmap et al, 2007) and DSB maps of PRDM9[A] carriers (Pratto et al, 2014) (Figs 1A, S1–S6). The estimated CO center lies in close proximity to an active PRDM9[A]-binding site (287 bp). The average NCO center is also very close to this central PRDM9-binding site (197 bp; Table S5); although the exact location of NCO centers is more ambiguous and based on fewer informative SNPs and smaller sample sizes. Three additional canonical PRDM9-binding motifs occur within our hotspot, but this central PRDM9[A]-binding site was described as the most active one (Altemose et al, 2017). Thus, we assume that most DSBs occur at or in close proximity to this PRDM9 motif (highlighted throughout the figures) and that the repair and resolution of the DSB into COs or NCOs develops in the vicinity of the two long poly-A's (9A/19A and 23A).

Interestingly, H3K4me3 (a nucleosome mark observed in active hotspots), as measured in human spermatocytes (Pratto et al, 2014) and PRDM9[B]-transfected cells (Altemose et al, 2017), is lowest at the sites harboring the longer poly-A repeats (9A/19A and 23A), located ~−160 or ~+240 bp from the most active PRDM9-binding site, respectively (Fig 1C). This pattern is unusual: typically, H3K4me3 at hotspots is lowest at the PRDM9-binding site and highest 150–250 bp upstream and downstream from it (Baker et al, 2014; Lange et al, 2016; Powers et al, 2016) and (reviewed in Paigen & Petkov (2018)).

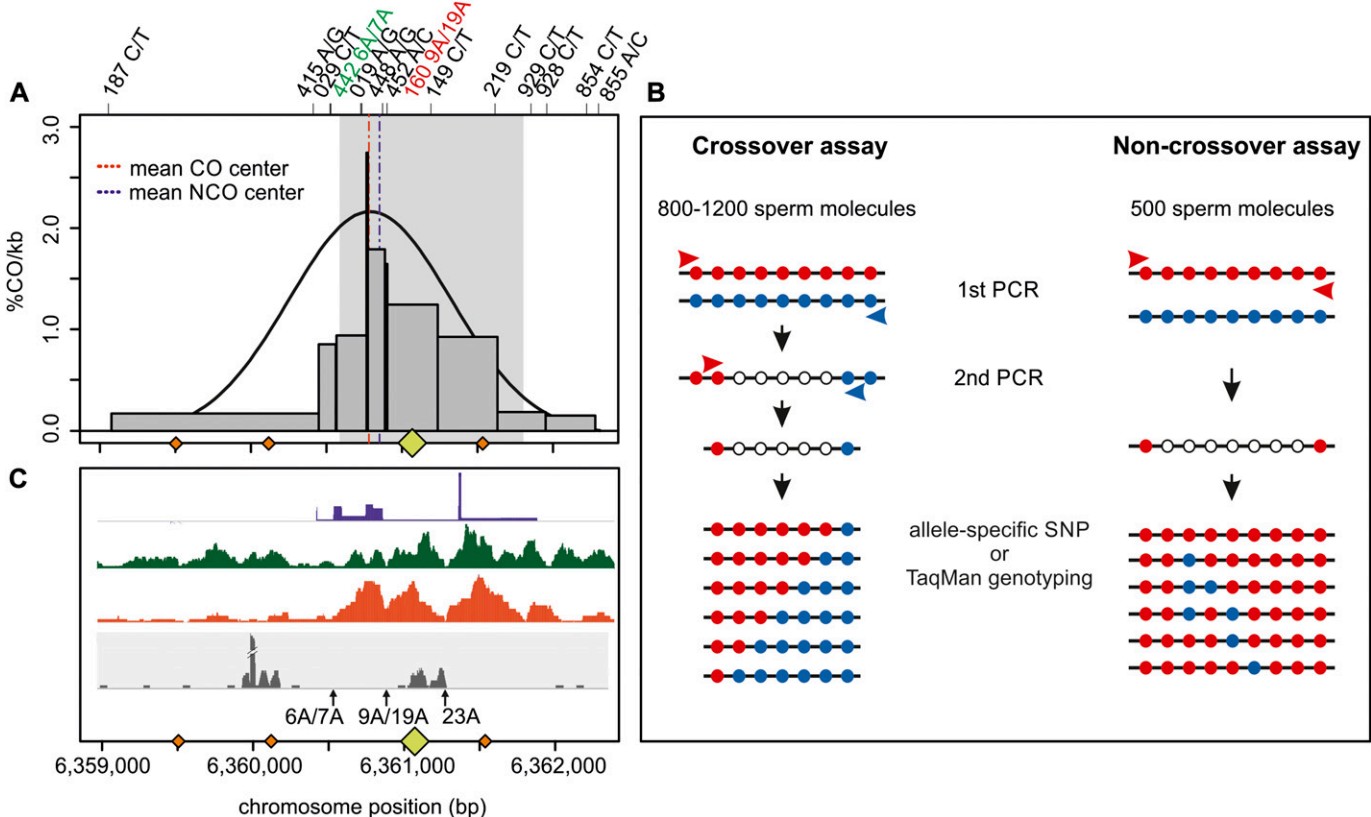

**Figure 1. Features and analysis of HSII.**
**(A)** Distribution of CO breakpoints (grey bars) measured with pooled-sperm typing in eight different donors. The mean CO and NCO centers (dashed lines) were estimated at chr16: 6,360,770±9 bp and 6,360,860±15 bp (GRCh37/hg19), respectively (Figs S1–S6 and Table S2). Orange rhomboids on the x-axis represent the PRDM9[A]-binding motif with up to one mismatch (CCnCCnTnnCCnC, where n reflects any base A, C, T, G with the same likelihood) (Myers et al, 2008). The larger yellow rhomboid at position chr16: 6,361,057–6,361,088 is likely the most active motif (verified to bind PRDM9 in transfected cells with a significant FIMO score; personal communication and (Altemose et al, 2017). The grey-shaded area represents the DSB region measured in spermatocytes (Pratto et al, 2014). **(B)** Graphical representation of the pooled-sperm typing assay to collect COs and NCOs. Approximately 800–1,200 or 500 sperm molecules were aliquoted per reaction for collecting COs or NCOs, respectively. COs were amplified with allele-specific primers with a perfect match at the 3′ end to the allele of the recombinant phase (red and blue arrows). The two nested PCRs produced mainly crossover amplicons. The NCO assay used allele-specific primers to amplify only one of the parental homologues. The phase switch of internal alleles representing the NCO was assessed by allele-specific PCRs targeting one SNP at a time. **(C)** Additional features of HSII as described in Altemose et al (2017). The first lane represents the historical recombination map inferred with LDhat (International Hapmap et al, 2007) in dark blue, the second lane is the measured H3K4me3 in human spermatocytes of PRDM9[A] carriers (Pratto et al, 2014) in green, and the third lane represents the H3K4me3 sites measured in HEK293T cells transfected with PRDM9[B] in bright red (Altemose et al, 2017). The grey panel plots the transcripts per million from permanganate/S1 footprinting for single-strand DNA (ssDNA) and non-B DNA sequencing (Kouzine et al, 2017) representing structures flanking non-B DNA. The black arrows denote the location of the three poly-A sites within HSII.

Although we cannot exclude the possibility that low H3K4me3 is linked to differences in coverage in difficult-to-sequence regions, it is also possible that the observed H3K4me3 patterns in our hotspot are correlated with the effect of poly-A's on nucleosome occupancy. Poly-A tracts resist nucleosome binding, which requires sharp bends in DNA that destroy purine–purine stacking and the zig-zag structure of additional non–Watson–Crick hydrogen bonds (Nelson et al, 1987). As a result, poly-A repeats show poor nucleosome occupancy, with longer repeats showing more extreme depletion (Field et al, 2008; Segal & Widom, 2009). The two longer poly-A runs (9A/19A and 23A) in our hotspot probably have low affinities for nucleosomes and thus reduced levels of H3K4me3. It is also possible that the open chromatin structure resulting from nucleosome depletion at these sites makes the PRDM9 motif closest to the poly-A's more accessible for binding than the other nearby motifs. To further underline this observation, we retrieved

permanganate/S1 footprinting data that detect single-stranded DNA and non-B DNA structures in the genome, which show a strong inverse correlation with nucleosome occupancy (Kouzine et al, 2017). We observed an increase in reads flanking the 9A/19A poly-A and 23A STR (Fig 1C, grey panel), suggestive of non-B DNA structures associated with low nucleosome occupancy. Interestingly, we also observed a second peak in our region overlapping a PRDM9 motif somewhat upstream that we cannot explain, exclusively.

## Long heterology at the 9A/19A STR influences the recombination outcome of CO or NCO

We estimated a mean recombination frequency of 79 cM/Mb at HSII, similar to the HapMap value of ~63 cM/Mb obtained from linkage disequilibrium data (International Hapmap et al, 2007). We asked

whether the central asymmetric 9A/19A repeat influences DSB repair and the outcome into CO or NCO. We found that outcomes differ significantly between 9A/19A Ht and 19A Ho donor groups (Fig 2 and Table S2). There was an approximately twofold decrease in CO events in the 9A/19A Ht versus 19A Ho group (exact two-sided Poisson test, EPT, Ht:Ho rate ratio 0.6 ± 95% CI [0.6–0.7], $P < 2.2 \times 10^{-16}$; Fig 2A). Conversely, NCO events in Ht versus Ho donors increased by roughly twofold to threefold (Ht:Ho rate-ratio 2.2 ± [1.7–2.9], $P < 3 \times 10^{-10}$). Furthermore, the overall recombination frequency (sum of CO and NCO) is roughly the same among donor groups (Fig 2B).

## The long heterology reduces GC-biased gene conversion

We next examined the segregation of alleles by comparing recombination products for both reciprocals. According to Mendel's law of equal segregation, on average, the alleles on each homologue are transmitted equally. However, during recombination, heterozygous sites of paired homologues form a heteroduplex; repair of these heteroduplex sites results in gene conversions in which one of the two alleles is removed. If the repair is biased, this process can lead to a higher transmission frequency of one allele over the other. Such biased repair underlies GC-biased gene conversion (gBGC): the over-transmission of "strong" G/C over "weak" A/T alleles, observed at hotspots in several species (reviewed in (Duret and Galtier, 2009; Lassalle et al, 2015; Tiemann-Boege et al, 2017)).

Here, we analyzed gBGC by comparing the reciprocal alleles proximal and distal to the CO breakpoints of all collected COs in a contingency table analysis. Specifically, we used a Cochran–Mantel–Haenszel framework to quantify the transmission advantage of heterozygous alleles as described previously (Arbeithuber et al, 2015). We calculated a significant excess of COs with strong alleles (rate-ratio of GC to AT alleles = 1.27; $P < 1 \times 10^{-4}$), implying GC alleles are transmitted to 53% in sperm with a CO or 50.013% of all

sperm in the tested donors (Table 1). The resulting transmission advantage due to gBGC ($b$) of $2.7 \times 10^{-4}$ is similar to that reported in another hotspot ($4.6 \times 10^{-4}$) (Arbeithuber et al, 2015).

Interestingly, gBGC appears impeded by asymmetry at the 9A/19A STR: the Ho donor group had significantly stronger gBGC than the Ht group (Table 1; $RR_{Ho} = 1.5$, versus $RR_{Ht} = 1.1$, respectively). For NCOs, as well, gBGC is found in the Ho group, but not in the Ht group. In fact, NCOs in the Ht group show a bias favoring weak alleles (Table 1). Inspection of the allelic transmission at individual polymorphic sites in CO (logRR plots in Fig 3B and Figs S7–S12B) suggests an explanation for this difference in gBGC between donor groups in CO: the 19A homologue and its flanking alleles are also over-transmitted; as 19A is mainly in phase with A/T alleles, this effect overpowers the overall gBGC.

## Transmission of STRs is influenced by the length of the heterology

Next, we examined the allelic transmission of the two heterozygous STRs (9A/19A or 6A/7A). Overall, we observed that the longer allele was over-transmitted in COs at both STRs as shown in Table 1 and Figs 3B, S7–S12B. Note that iBGC in COs is mainly driven by the 6A/7A STR, which showed a strong and significant iBGC (RR = 1.95; $P$-value < $1 \times 10^{-4}$), whereas the asymmetric 9A/19A STR was not biased and significant (RR = 1.02; $P$-value = 0.07). Interestingly, the Ht donors had better support for iBGC at the 6A/7A STR site than the Ho group (RR = 1.95 versus 1.56, respectively).

For NCO, the transmission patterns support also the preference for the longer allele at the 6A/7A STR, despite the scarcity of conversions here (Table 1). However, for the 9A/19A in NCO, the trend is reversed, with the 9A being transmitted significantly more often than the longer 19A (RR = 0, $P < 10^{-6}$) in all types of conversion events (simple NCOs, co-conversion, and complex conversion events). Overall, averaging over both COs and NCOs, we find insertion-biased transmission at the 6A/7A site (recovered in 57.1% of NCO and CO molecules, using FxR from Table 1 and the CO:NCO

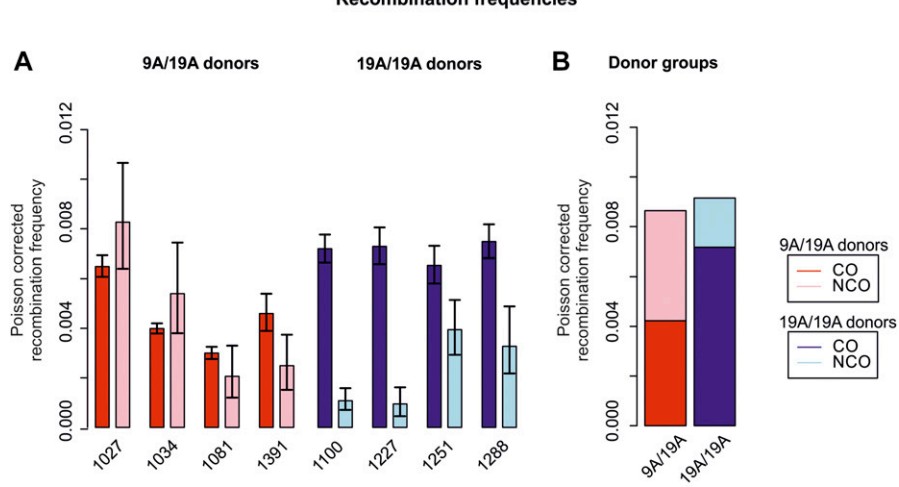

**Figure 2. Recombination frequencies of CO and NCO measured in HSII.**
**(A)** CO and NCO frequencies compared by individual donors and donor groups. CO frequencies (red) of 9A/19A heterozygous (Ht) donors are lower than CO frequencies of homozygous (Ho) donors (dark blue). This trend is reversed for NCOs, in which NCOs are more frequent in Ht (light red) than in Ho (light blue) donors. Error bars denote confidence intervals calculated by an exact two-sided Poisson test. **(B)** Average CO and NCO frequency in Ht and Ho donor groups.

**Table 1. Estimated gBGC and iBGC for CO and NCO grouped into all donors (Ht+Ho), Ht, or Ho.**

| | Rate Ratio-RR | *P*-value | FxR | *c* | Fx total (HSII) | b |
|---|---|---|---|---|---|---|
| *gBGC* | (strong/weak) | | | | | |
| CO | | | | | | |
| Ht+Ho | 1.27 (1.17–1.38) | $<1 \times 10^{-4}$ | 53.0% | $2.5 \times 10^{-3}$ | 50.013% | $2.68 \times 10^{-4}$ |
| Ht | 1.10 (0.98–1.23) | 0.13 | 51.2% | $2.1 \times 10^{-3}$ | 50.004% | $8.76 \times 10^{-5}$ |
| Ho | 1.50 (1.33–1.69) | $<1 \times 10^{-4}$ | 55.1% | $3.6 \times 10^{-3}$ | 50.031% | $6.28 \times 10^{-4}$ |
| NCO | | | | | | |
| Ht+Ho | 0.93 (0.73–1.19) | 0.58 | 49.0% | $7.1 \times 10^{-4}$ | 49.999% | $-1.41 \times 10^{-5}$ |
| Ht | 0.61 (0.42–0.86) | $<1 \times 10^{-2}$ | 43.9% | $1.1 \times 10^{-3}$ | 49.993% | $-1.35 \times 10^{-4}$ |
| Ho | 1.54 (1.06–2.26) | 0.02 | 55.4% | $5.0 \times 10^{-4}$ | 50.003% | $5.38 \times 10^{-5}$ |
| *iBGC* | (insertion/deletion) | | | | | |
| CO | | | | | | |
| Ht 6/7 +9/19, Ho 6/7 | 1.17 (1.12–1.23) | $<1 \times 10^{-4}$ | 52.0% | $2.1 \times 10^{-3}$ | 50.004% | $8.27 \times 10^{-5}$ |
| Ht+Ho 6/7 | 1.91 (1.60–2.28) | $<1 \times 10^{-4}$ | 58.0% | $3.1 \times 10^{-3}$ | 50.025% | $4.97 \times 10^{-4}$ |
| Ht 6/7 + 9/19 | 1.17 (1.12–1.22) | $<1 \times 10^{-4}$ | 52.0% | $2.9 \times 10^{-3}$ | 50.006% | $1.14 \times 10^{-4}$ |
| Ht 9/19 | 1.02 (1.00–1.04) | 0.07 | 50.2% | $2.1 \times 10^{-3}$ | 50.001% | $1.05 \times 10^{-5}$ |
| Ht 6/7 | 1.95 (1.62–2.34) | $<1 \times 10^{-4}$ | 58.3% | $2.9 \times 10^{-3}$ | 50.024% | $4.80 \times 10^{-4}$ |
| Ho 6/7 | 1.56 (0.80–3.03) | 0.25 | 57.2% | $3.6 \times 10^{-3}$ | 50.026% | $5.18 \times 10^{-4}$ |
| NCO | | | | | | |
| Ht 6/7 +9/19, Ho 6/7 | 0.59 (0.32–1.05) | 0.08 | 43.4% | $1.4 \times 10^{-3}$ | 49.991% | $-1.87 \times 10^{-4}$ |
| Ht+Ho 6/7 | 1.54 (0.73–3.37) | 0.30 | 55.4% | $7.2 \times 10^{-4}$ | 50.004% | $7.36 \times 10^{-5}$ |
| Ht 6/7 + 9/19 | 0.25 (0.08–0.63) | 0.001 | 33.3% | $1.3 \times 10^{-4}$ | 49.978% | $4.33 \times 10^{-4}$ |
| Ht 6/7 | 2.0 (0.4–12.3) | 0.51 | 58.6% | $1.3 \times 10^{-3}$ | 50.011% | $2.23 \times 10^{-4}$ |
| Ht 9/19 | 0.0 (0.00–0.19) | $<1 \times 10^{-6}$ | 0.0% | $1.1 \times 10^{-3}$ | 49.945% | $-1.10 \times 10^{-3}$ |
| Ho 6/7 | 1.4 (0.58–3.52) | 0.54 | 54.2% | $5.0 \times 10^{-4}$ | 50.002% | $4.20 \times 10^{-5}$ |

RR is the rate ratio of strong to weak alleles for gBGC or long versus short alleles for iBGC and approximates the value of one when alleles are equally transmitted; *P* is the significance of the RR; FxR is the percentage of recombinants with a biased transmission estimated as $\sqrt{RR}/(1+\sqrt{RR})$, where 50% represents equal transmission; and *c* is the number recombinant products per amplifiable sperm genomes (see Table S2). Note that we used 2*CO, because two crossovers result per meiosis, yet we only measured one type (RI or RII) and 1*NCO, because only one NCO results in a meiotic division (Allers & Lichten, 2001); Fx total is the transmission bias in all sperm estimated as the sum of the proportion of non-recombinants and proportion of FxR (calculated as: 0.5*(1–FxR)+*c**FxR); b is the selection coefficient estimated as (2*Fx total)–1.

ratio from Table S2), whereas, we find deletion-biased transmission at the 9A/19A site (32% of NCO and CO molecules; Materials and Methods [Data analysis] section of the Supplementary Information).

**The 9A/19A STR reduces CO exchanges and increases complex conversions**

The repair of meiotic DSBs is expected to result in a new arrangement of phased alleles in both COs and NCOs. For both COs and NCOs, recombination breakpoints are expected to accumulate near the DSB site, with the exact breakpoint determined by the extent of repair near the DSB (Jeffreys et al, 2001; Jeffreys and May 2004; Odenthal-Hesse et al, 2014; Tiemann-Boege et al, 2006; Arbeithuber et al, 2015). The exchange points in our hotspot (Figs 3, S7–S12) also show this pattern, except for the asymmetric 9A/19A STR in Ht donors. The Ht donor group shows a unique pattern with reduced or absent CO exchange points directly at the 9A/19A in the middle of the hotspot (Figs 3A, S7A, and S8A). We observed this pattern for all three Ht donors that were informative for

several closely spaced SNPs in this region (three out of four Ht donors). Such "gaps" in CO breakpoints have been previously observed at positions of palindromic repeats, and long and complex micro- or even minisatellites with inversions in mice (Baudat & De Massy, 2007; Cole et al, 2010; Wu et al, 2010) and humans (Jeffreys & Neumann, 2005), but not in the context of an asymmetric mononucleotide run. Here, this CO gap is instead caused by a relatively minor length polymorphism of 10 bp within a perfect poly-A mononucleotide repeat.

CO and NCO events mainly concentrate within the same region (Figs 3C, S7–S12). In NCO events, most conversions (~83%, Table S6) involve only one SNP with an estimated mean tract length of 1,037 ± 1,264 bp (Table S7) for all eight donors, with no difference in tract length between donor groups. Conversions involving a single SNP are not unusual and were initially observed in several mice hotspots, which had higher SNP densities than our donors (Cole et al, 2010), and have also been previously observed in humans (Odenthal-Hesse et al, 2014). Despite the lower density of informative SNPs in our hotspot, we identified ~4% of co-conversions

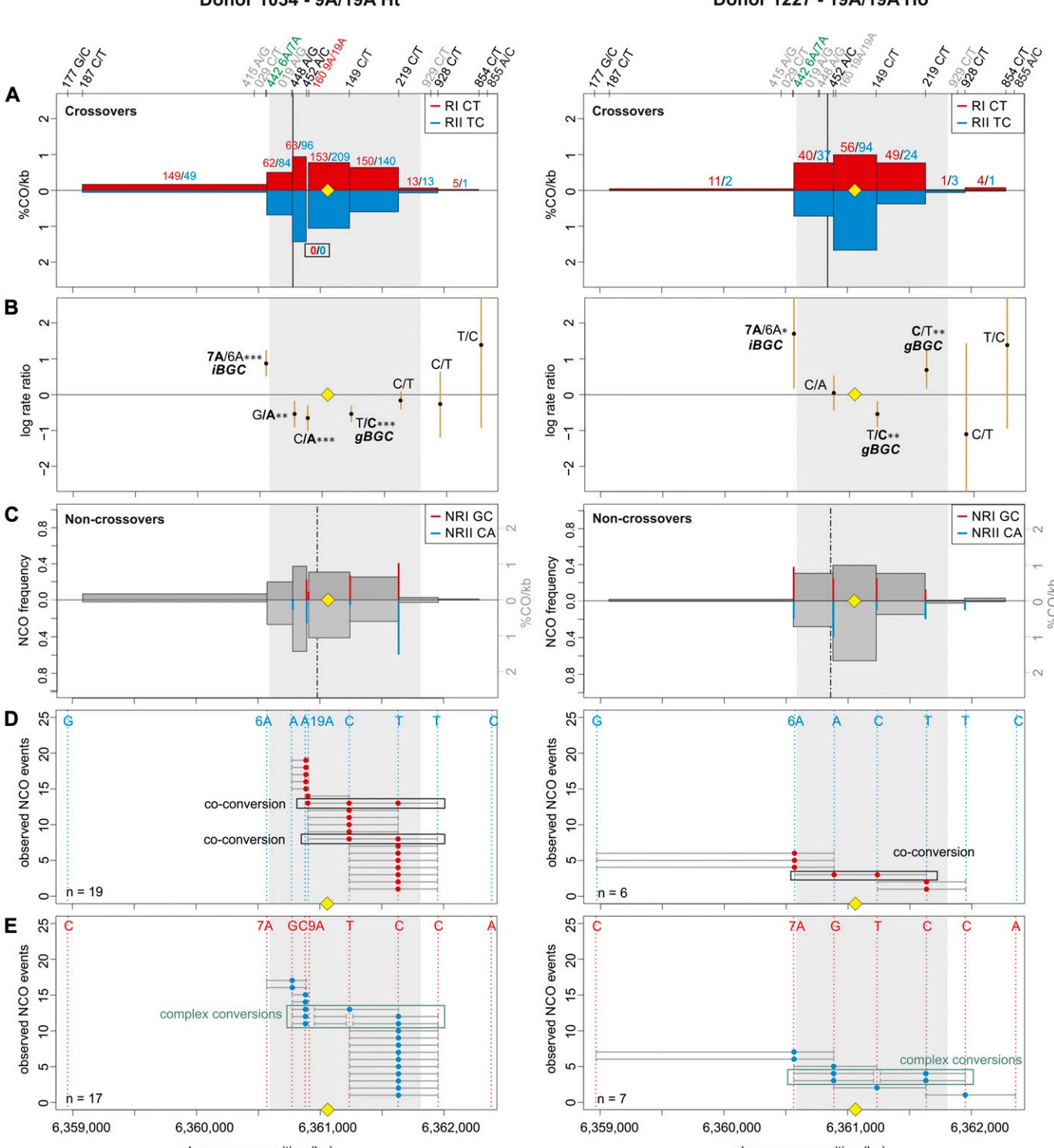

**Figure 3. CO and NCO transmission of 9A/19A Ht donor (left) and 19A/19A Ho donor (right).**

**(A)** CO transmission between reciprocals. CO breakpoint distributions of both reciprocal products based on n = 1,313 and 344 collected CO products for donor 1034 and donor 1227, respectively (also see Table S2). Note that numbers on top of the breakpoint sites are normalized to represent equal numbers of collected reciprocals. The average CO centers estimated either for the Ht or the Ho group is denoted by the black vertical lines, the grey area denotes the DSB zone (Pratto et al, 2014), and the yellow rhomboid represents the PRDM9-binding site. Note the absence of breakpoints at the central 9A/19A STR for donor 1034. **(B)** Biased CO transmission. Transmission differences between the alleles of reciprocal COs estimated by the log rate ratio of the different recombinant haplotypes, calculated as log[(nRI/totalRI)/(nRII/totalRII)], where the denominator is the total number of normalized CO surveyed per reciprocal. The horizontal line at logRR = 0 denotes the expected equal transmission of alleles between the reciprocal recombinant haplotypes. Asterisks denote a significant over-transmission (logRR > 0) or under-transmission (logRR < 0) based on the standardized

involving two SNPs, but a few also spanned over three SNPs (~1%; Table S6).

We also observed complex COs (CCO) with unconverted SNPs between converted ones within 129–1,177 nt in both Ht and Ho donor groups occurring at similar frequencies (1.6% and 1.2% CCO/CO or $6.69 \times 10^{-5}$ and $8.66 \times 10^{-5}$ CCO/meiosis, respectively; Tables S8 and S9 and Fig S13). Interestingly, these complex conversion tracts occur within short distances compared with reported complex conversion tracts occurring over distances of 100 kb (Halldorsson et al, 2016) and ~12 kb (Williams et al, 2015). Approximately 70% of the complex tracts were located upstream of the 9A/19A STR or directly at this STR position and ~30% were located directly after the STR position (see Table S9). The frequency of CCO was ~3 times higher than that reported for other hotspots (CCO/CO = 0.35%, 0.21–0.54) (Arbeithuber et al, 2015) and 0.33% (Webb et al, 2008). Previously, Arbeithuber et al collected in parallel an extensive number of negative controls (one non-recombinant genome in a pool of sperm or blood genomes of another donor with the recombinant haplotype) for two different hotspots including HSII. Approximately 1.0% CCO/CO was observed for donor 1081, but no complex conversions were observed in the negative controls, verifying that these events are not the result of technical artifacts (see details in Arbeithuber et al (2015)).

Interestingly, we also observed that ~12% of the NCO involved complex conversions (Table S6) with tract lengths of 1,543 bp ± 793 bp that carried unconverted SNPs between converted ones, as seen in Figs 3D and E, S7, S9–S11D, and E. We observed complex NCOs in six of eight donors. In most complex conversions, only a single SNP was unconverted (Figs 3D and E, S7, S9, S10–S11D, and E), which involved in all cases the 9A/19A STR and/or flanking SNPs to this STR (SNP rs149 C/T). Of note is the significant threefold higher frequency of these complex conversions in the Ht over the Ho donor group per amplifiable sperm, in which the 9A is the unconverted allele (Fig S13 and Table S8). These complex conversions explain the highly significant deletion bias observed at the 9A/19A (RR = 0.00; $P < 1 \times 10^{-6}$), favoring the transmission of the 9A over the 19A (Table 1).

### Poly-A's are enriched in recombination hotspots but are not more diverse

Given the strong iBGC in our hotspot for the short poly-A (6A/7A), we examined genomic data for signatures of enrichment of poly-A's at hotspot locations using the R-package STRAH (see https://github.com/PhHermann/STRAH). Specifically, we retrieved the genomic coordinates of all poly-A's with perfect repeat tracts of six or more A's from the reference genome (GRCh37/hg19). We then compared the enrichment of poly-A's in recombination hotspots to hotspot-flanking regions of 1-kb sliding windows (five windows in total). Given that recombination is limited to a small proportion (~5%) of the human genome (Myers et al, 2005), we limited our

comparisons to hotspots and flanking regions, given their similar genomic contexts subject also to other biological mechanisms acting at a broad scale that could drive poly-A enrichment. Fig 4A shows that hotspots are enriched for poly-A's (total number of poly-A's counted in each zone within the corresponding zone length) compared to flanking regions. We observed a significant ~twofold increase in poly-A's and density of A's within a repeat in hotspots compared with flanking regions (Fig 4B).

In Fig 4C and D, we compared the length of poly-A repeats inside hotspots and flanking regions and did not observe longer tracts within hotspots. In fact, the density of longer poly-A's in hotspots rapidly decayed with repeat length (Fig 4C), with 90% of the poly-A's being 6–11 nt in length (mean = 7.77, median = 6) in hotspots and flanking regions. Regardless of the length, poly-A's have ~twofold enrichment within hotspots compared to flanking regions (Fig 4D). Similar patterns were also observed for poly-T's (Fig S14).

We also tested for an effect of hotspots on the diversity of poly-A's, as would be expected if these poly-A's are more unstable than poly-A's more distant from a hotspot. For this analysis, we used the lobSTR reference sites from the Simons Genome Diversity Project (SGDP) (Mallick et al, 2016) that were variable in individuals of West Eurasian descend (likely PRDM9[A] carriers). Most poly-A's, whether in hotspots or flanking regions, had an average of three alleles per site, with only one nucleotide difference between repeats. We tested four estimates of diversity: heterozygosity, allelic asymmetry (difference between the longest and shortest allele), steps between alleles (unit differences between alleles), and the total number of different alleles in the population (see the Materials and Methods section for details). None of the diversity measures showed any difference between poly-A's in hotspots and flanking regions. Moreover, we observe no difference in asymmetry between major alleles in hotspots compared to flanking regions (Fig S15). Interestingly, in this data set, we also observed poly-A enrichment of 1.5- to 2-fold bordering on significance. Moreover, this enrichment decreased rapidly with tract length and was almost absent with poly-A's greater than 23 A's (Fig S16). Because lobSTRs were only called for poly-A's with 11 to ≥26 A's, we repeated our analysis with poly-A's from the genome reference restricted to the same lengths, showing also a significant twofold enrichment at hotspots as described in Fig 4 (Fig S17).

Given the strong iBGC and insertion mutation bias, a lack of increase in variability of poly-A's at hotspots is surprising. However, it is possible that a sample size larger than our tested SGDP population is required to obtain a measurable effect (with a transmission frequency of 50.004%, only seven sites are expected to expand in the 175,384 poly-A sites retrieved from the SGDP, see Table 1). In fact, a slight positive association was described previously when comparing heterozygosity with broad-scale recombination, yet considering all mononucleotides (Mallick et al, 2016).

Pearson residual. Three asterisks denote the strongest biased transmission ($P < 0.001$), and two and one asterisk represent a $P$-value of <0.01 and $P < 0.05$, respectively. **(C)** NCOs overlap with CO frequencies. Shown are NCO frequencies (Poisson corrected and normalized between reciprocals) as red and blue lines compared with CO frequencies as grey shaded areas from panel A, and the estimated NCO center averaged over Ho or Ht group as a black dashed line. **(D, E)** Observed NCOs for both reciprocals. Individual NCOs showing the converted alleles. The possible conversion tract length is denoted as a fine horizontal grey line between informative SNPs (shown on top of the panel). The mean conversion tract length is 625 bp and 1,354 bp for donor 1034 of donor 1227, respectively. Most NCOs are single conversions involving only one SNP; however, co-conversions (tracts with more than one converted allele) and complex conversions (conversion tracts with a mixture of converted and original parental alleles) also are observed.

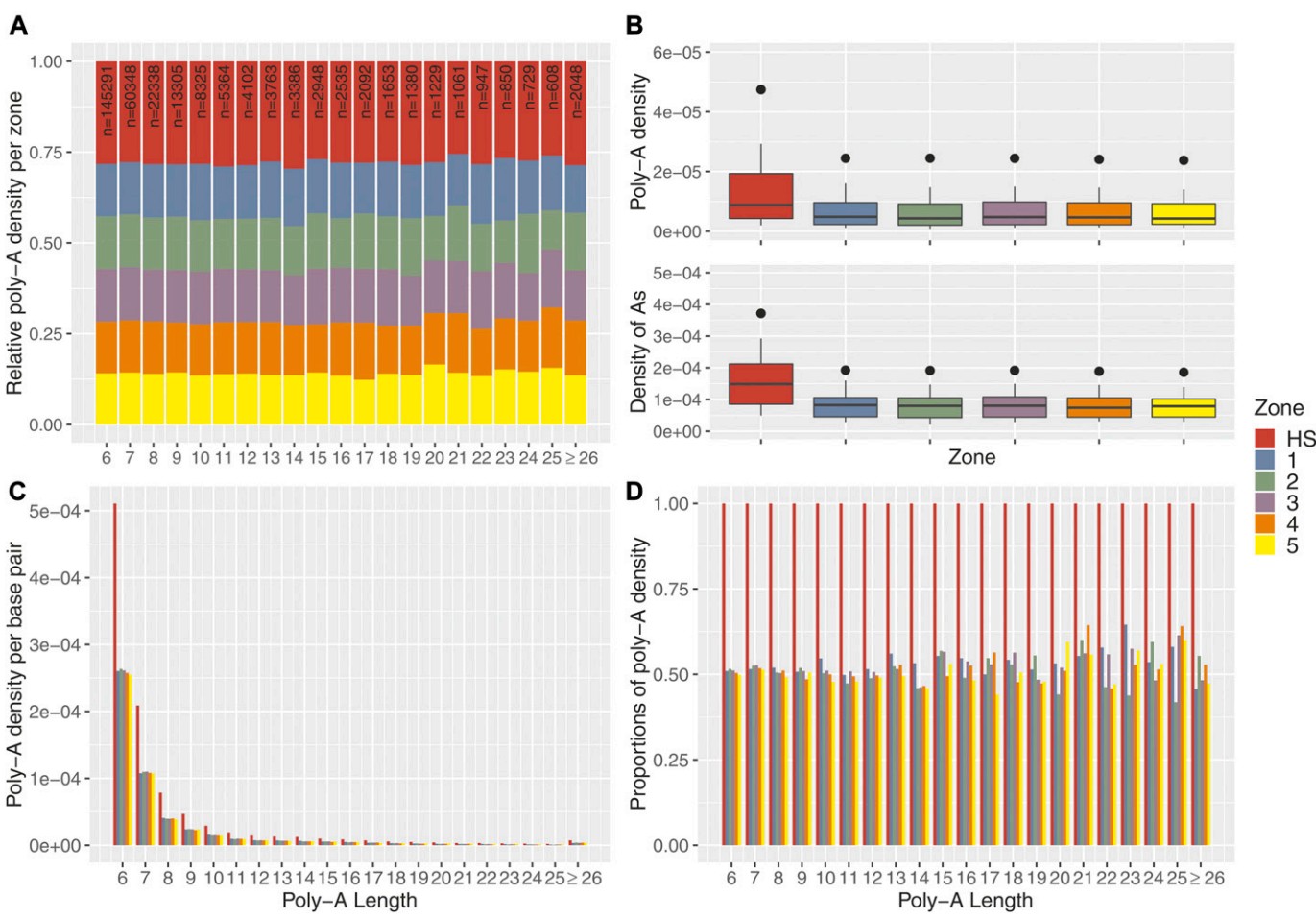

**Figure 4. Poly-A enrichment at recombination hotspots.**
**(A)** The poly-A density is the number of poly-A's divided by the zone length. For each poly-A tract length (6 to ≥ 26 A's), the densities were re-normalized [0,1] by the sum of all densities. We extracted poly-A's (total number of considered poly-A's, n = 284,302), from the reference genome (GRCh37/hg19) that fall either within hotspots (HS; red), within flanking regions (five sliding windows left and right of the hotspot, each 1 kb in length; 1–5). Hotspots were defined as ±500 bp from the DSB coordinates of PRDM9[A] carriers identified by (Pratto et al, 2014), leading to an average hotspot length of ~2 kb. The subsequent zones were chosen as 1-kb segments upstream and downstream from the boundaries of the hotspot (2 kb in total per zone). Note that repeats with at least 26 A's are pooled into one class. **(B)** Top panel: the poly-A density is the number of poly-A's in a zone divided by the length of this zone in base pairs. Bottom panel: the densities of A's per zone are calculated by dividing the number of A's (length of the poly-A times its frequency) by the length of the zone in base pairs. The enrichment of poly-A's within the hotspot compared with the flanking regions is approximately twofold for the poly-A densities and for the densities of A's (in terms of mean and median). A Kruskal–Wallis test comparing all poly-A's in hotspots versus all flanking regions leads to highly significant results ($P < 1 \times 10^{-3}$ and $P < 1 \times 10^{-5}$ for poly-A density or density of A's, respectively). **(C)** The poly-A densities per base pair are shown stratified with respect to the length of the poly-A tract. **(D)** The densities in the flanking regions are displayed as fractions relative to the densities within hotspots to better distinguish the enrichment for longer poly-T tracts.

# Discussion

### STR length heterology influences DSB repair

Meiotic recombination is initiated by programmed DSBs via SPO11, followed by a series of repair steps from strand resection to invasion, the formation of complex intermediate structures between homologs, and the repair of mismatches of heteroduplexes in paired homologue strands (Fig 5). The intermediate structures can be repaired either by double-strand break resolution, leading mainly to COs and NCOs, or by synthesis-dependent strand annealing, resulting in NCOs (reviewed in (Arnheim et al, 2007; Paigen and Petkov, 2010; De Massy, 2013; Lam and Keeney, 2014; Tiemann-Boege et al, 2017)).

In our analysis of thousands of CO and NCO products from two different donor groups (9A/19A Ht and 19A/19A Ho), we observed that a short heterology (6A/7A) leads to a transmission bias of the longer allele, whereas a heterology of 10 nt leads to more transmissions of the shorter allele. Moreover, the length asymmetry between STRs also has an effect in the recombination outcome: the 9A/19A heterology at the center of the hotspot and in close proximity to the DSB site (~160 bp) reduces the number of DSBs repaired as COs that are alternatively repaired as NCOs. Also, the 9A/19A asymmetry slightly reduced the overall recombination frequency and resulted in other events such as larger conversion tracts leading to reduced gBGC, central gaps in CO exchanges at the asymmetric site, and a higher frequency of complex conversion events in NCOs.

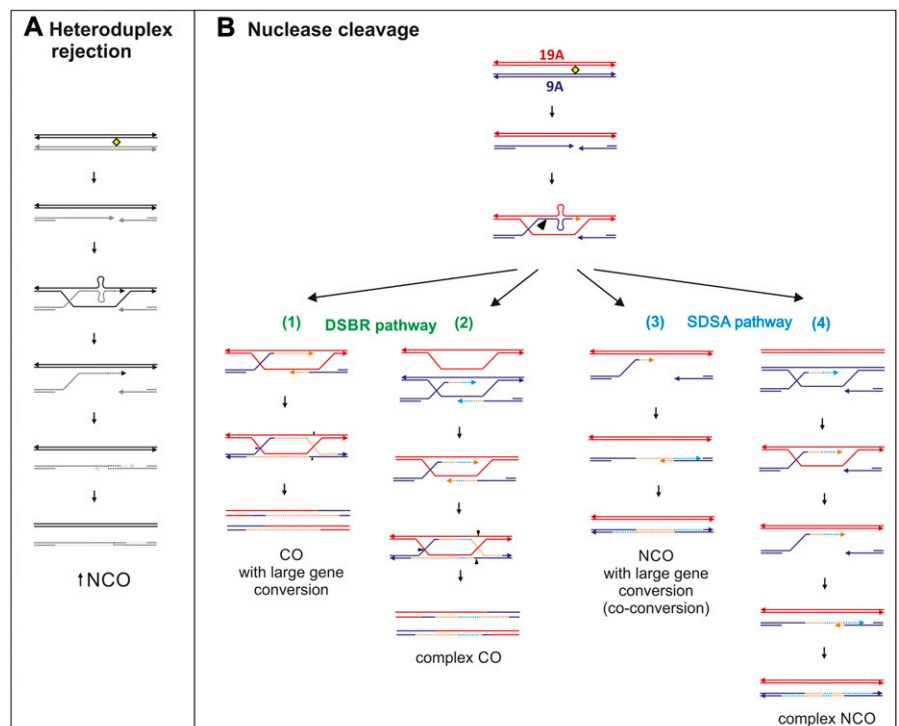

**Figure 5. DSB repair of central 9A/19A repeat.**
**(A)** The 9A/19A asymmetry can destabilize strand invasion leading to subsequent heteroduplex rejection, resulting in more NCOs via the synthesis dependent strand annealing pathway. **(B)** The formation of a 10-bp heteroduplex activates the mismatch repair (MMR)/large loop repair (LLR) system which likely removes the heterology by nuclease cleavage creating a large double-strand gap. Double-strand break repair (DSBR) forms COs or NCOs with large conversion tracts (1). Note that DSBR can also result in NCOs depending on the double Holliday cleavage sites (not indicated in the figure). Orange dashed lines represent the newly synthesized DNA. In case of sister-strand invasion (blue dashed lines), complex conversion tracts are formed (2). If strand displacement happens after the MMR endonucleolytic digestion past the asymmetry, NCOs with large conversion tracts (co-conversions) are formed (3), or alternatively complex conversions retaining the 9A allele via inter-sister repair or possibly by LLR (not indicated) (4).

Why a length asymmetry in STRs shifts the fate of DSBs and influences its own exchange and that of flanking SNPs is not known, but it could be linked to the formation of DNA loops at the heteroduplex 9A/19A STR site (Fig 5). These structures are likely temporary slipped structures, as are predicted for misaligned repeats (reviewed in Wang & Vasquez (2006)). Similar outcomes have been reported for *stable* loops at recombination hotspots: reduced exchange between homologs at the loop site, so called "exchange gaps" were observed in the context of inverted repeat structures of ~140 bp (Cole et al, 2010), indels of 20–50 bp in mice (Baudat & De Massy, 2007; Wu et al, 2010; Paigen et al, 2008; Bois, 2007), at indel positions in *Arabidopsis thaliana* (Drouaud et al, 2013), and at the palindromic AT-rich minisatellite MSNID in humans (Jeffreys & Neumann, 2005). Two of these studies also reported an increased NCO frequency at these sites, similar to that seen here (Jeffreys & Neumann, 2005; Cole et al, 2010).

Note that unlike the structures in these previous studies, which form large and stable loops (e.g., hairpin loops or cruciform structures, in the case of the inverted repeat), simple poly-A's do not form stable secondary structures (Snyder et al, 2008). It is thus surprising that a 10-bp mismatch is sufficient to trigger a similar exchange gap. Large, stable structures can impair or even prevent strand invasion and exchanges within the minisatellite (Jeffreys & Neumann, 2005) or also attract nucleases that remove the loop and produce longer conversion tracts (co-conversions) (Cole et al, 2010). In addition, inverted repeats or palindromes capable of forming stable hairpins or cruciform structures at recombination hotspots have been reported to result in promiscuous, SPO11-independent, DSBs and large, deleterious chromosomal rearrangements (e.g., non-Robertsonian translocations reviewed in Richard et al (2008)).

## How are heterozygous STRs repaired?

Although we cannot extrapolate the expected trends for all levels of asymmetry or types of STRs from our data, our results suggest that length asymmetries (e.g., 9A/19A) forming temporary loop structures trigger a series of alternative repair pathways.

Single nucleotides or 1–2-bp indels are recognized by the MSH2–MSH6 complex, and longer indel loops attract MSH2–MSH3 of the mismatch repair (MMR) system (reviewed in Spies & Fishel (2015)) and yeast (reviewed in Chakraborty & Alani (2016), Manhart & Alani (2016)). The MMR complex initiates repair in 5' or 3' direction starting from a nick or strand break (reviewed in Jiricny (2006)), with the efficiency of repair decreasing proportionally with increasing loop sizes (Jensen et al, 2005, Mcculloch et al, 2003a).

For larger loops, an alternative repair system, known as large loop repair (LLR), is likely active instead of MMR. LLR activity has been described for MMR-deficient cells in yeast (Kirkpatrick & Petes, 1997; Corrette-Bennett et al, 2001) and human cells (Mcculloch et al, 2003b). LLR starts the repair either from a nick or a gap in the 5'- or 3'- direction (nick-directed repair), similar to MMR, or simply removes the loop via endonucleolytic cleavage (loop-directed repair), regardless if the loop is located at a nicked strand or not (Mcculloch et al, 2003b). Loop deletion or retention depends mainly on the length and secondary structure of the loops (Bill et al, 2001). Short palindromic loops are preferentially retained (12 and 14 bp loops), whereas longer palindromic loops (40 bp) are removed (Bill et al, 2001). In comparison, non-palindromic, unstructured loops (e.g., temporary loops like our asymmetric poly-A) are preferentially removed independent of their size (Bill et al,

2001). The current model proposes that small loops (2–16 bp) are repaired by both LLR and MMR, but as loops get larger, LLR is the most prominent mechanism (Corrette-Bennett et al, 2001; Mcculloch et al, 2003a; Jensen et al, 2005; Sommer et al, 2008).

It cannot exactly be elucidated from our data which of the repair processes are acting on the temporary loops at our asymmetric 9A/19A STR, but certain trends can be recognized. It is possible that both homologues are equally targeted for DSBs based on the absence of PRDM9 motif differences between homologues and equal frequency of observed reciprocal NCOs (Table S4). However, we cannot discard the possibility that the homologue carrying the shorter poly-A (9A) is the preferred target for DSB given potential differences in nucleosome occupancy. In fact, the over-transmission of SNPs located ~130–560-bp upstream of the STR linked to the 19A homologue can be best explained by a scenario that starts with a DSB in the 9A homologue. The mismatch at the STR is then repaired either by LLR or MMR, directed by introducing a second 5′–3′ nick, which results in a larger DSB gap, as it was also suggested in a mouse hotspot for an inverted repeat forming a stable secondary loop (Cole et al, 2010). In contrast, DSBs in the 19A homolog are likely repaired only by LLR, which removes the loop, resulting in no over-transmission of 5′ flanking SNPs because longer loops >16 bp are processed only in 3′→5′ direction (Mcculloch et al, 2003a).

Support for active LLR in our hotspot also comes from our observation of complex conversions mostly carrying the short 9A allele, likely the result of more loop deletions than retentions, as proposed for non-palindromic structures (Bill et al, 2001). Furthermore, the fact that complex crossovers mainly occurred upstream of the 9A/19A STR supports an active LLR. However, we cannot exclude that inter-sister repair instead resulted in these complex conversions, as observed in (Tsaponina and Haber, 2014; Williams et al, 2015); Halldorsson et al, 2016). Although unusual, evidence of inter-sister repair of DSBs was shown in 13–25% of repair tracts in *Saccharomyces cerevisiae* (Schwacha and Kleckner, 1994, 1997; Oh et al, 2007; Jessop & Lichten, 2008). Moreover, CAG tracts and inverted repeats induce inter-sister recombination in yeast (Nag et al, 2004). Especially in regions of high heterozygosity (e.g., indels), the repair of DSBs via the sister, in addition to the homolog ensures timely DSB repair (Goldfarb & Lichten, 2010). The overall lower total recombination frequency observed in Ht donors could also indicate possibly active inter-sister repair. Alternatively, other pathways such as micro-homology–mediated end-joining or single-strand annealing leading to deletions could also explain our NCO data (reviewed in Polleys et al (2017)).

### Small heterologies lead to an insertion bias and large heterologies result in a deletion bias

The analysis of STR mutations in human pedigrees suggests that STR instability models should incorporate length asymmetry and heterozygosity (Amos et al, 2015). Our results support this view and clearly suggest that factors such as length asymmetry, heterozygosity, and the location of the poly-A's at recombination hotspots play an important role in their transmission and evolution. The heterozygosity and asymmetry at the 9A/19A at the hotspot center supports a deletion bias (over-transmission of the shorter allele),

whereas the heterozygous 6A/7A STR with only one mismatch located ~200 bp from the hotspot center supports an insertion bias (the longer allele is over-transmitted; Table 1). The transmission of the 9A/19A is subject to LLR processing, resulting in recombination outcomes favoring the shorter allele, especially in complex NCO. In contrast, the outcome of repair at the 6A/7A site is likely the result of MMR or possibly base excision repair (BER) acting on heteroduplexes formed in later steps of meiotic DSB repair, as proposed also for gBGC in yeast (Lesecque et al, 2013) and in humans (Glemin et al, 2015). MMR preferentially replaces the shorter for the longer allele in *Drosophila* (Ometto et al, 2005; Presgraves, 2006; Leushkin & Bazykin, 2013), as we also observed for the 6A/7A STR. Interestingly, the observed iBGC at the 6A/7A site is larger than our estimated gBGC for strong versus weak SNPs in our hotspot (RR = 1.9 versus 1.3, respectively).

A deletion bias was reported for COs and NCOs derived from a very large pedigree study (Halldorsson et al, 2016) and also for nonallelic gene conversions in *Drosophila* whole genome sequence comparisons (Assis & Kondrashov, 2012). A study that stratified indels into different lengths also observed a transmission bias favoring deletions associated with recombination in humans, flies, and yeast (Leushkin & Bazykin, 2013). However, for small indels (1–4 bp), the trend was reversed, with insertions favored instead, with the strongest bias measured for indels with 1-bp difference (approximately fivefold higher insertion over deletion rate; Leushkin & Bazykin, 2013). Although Leushkin and Bazykin (2013) measured an indirect statistical association between indel mutation rates and recombination rates in different organisms, which have recombination maps of varying resolution (Mb to bp), their findings are consistent with our model proposing that at recombination hotspots, STRs with small heterologies expand, whereas STRs with large heterologies contract. In a broader context, repair at heterozygous STRs requires an extra round of DNA synthesis of longer DNA tracts to remove the slipped DNA at the heteroduplex, which may introduce new mutations by error-prone polymerases active in MMR, explaining the observation of increased single-nucleotide changes flanking indels (Tian et al, 2008).

### Seeking for the hidden—poly-A enrichment at human recombination hotspots

Given the strong iBGC in our hotspot for the short poly-A (6A/7A), we predict that over evolutionary time, hotspots become enriched with poly-A's. Nucleotide composition at evolutionary equilibrium (described by the Li-Bulmer Equation $(1)/[1 + \kappa(\exp(-2N_e b)]$, Bulmer, 1991; Nagylaki, 1983) can be estimated considering a heterozygous selection coefficient favoring insertion over deletions (equivalent to the iBGC, $b$), the effective population size $N_e$, and the ratio of the opposing mutation rates, $\kappa$ (in/del versus del/in). Because there are no published indel mutation rates for individual poly-A lengths in humans, we extracted these data from a pedigree analysis (Fungtammasan et al, 2015). For repeats between 6 and 12 A's, the insertion rate is more frequent than the deletion rate (Table S10) (Sun et al, 2012), but this trend is reversed for longer poly-A's (>12 A's). Given that indel mutational bias changes with repeat length, the transmission pattern depending on heterozygosity and repeat asymmetry makes the prediction on the outcome of poly-A

evolution at hotspots rather complex. For short repeats, however, a back-of-an-envelope calculation shows that strong iBGC combined with a mutation bias favoring insertion predicts an enrichment of short-to-intermediate–sized poly-A's (<12 bp) at hotspots, consistent with the observed enrichment of poly-A's at hotspots compared with flanking regions. Similar findings from an analysis of STRs other than mononucleotide repeats in yeast also show a twofold enrichment of STRs near DSBs and recombination hotspots. The biological mechanisms driving this enrichment are unlikely to be mutagenesis, given the lack of de novo STR mutations measured at hotspots (Zavodna et al, 2018).

However, it could be possible that the cause of the observed poly-A enrichment at hotspots is due to inherent DNA properties of poly-A's promoting the formation of recombination hotspots and not the consequence of iBGC and insertional mutation bias. That is, as poly-A's expand, they lose nucleosome occupancy, creating open chromatin regions that extend beyond the poly-A tract by at least 100 bp (reviewed in Segal & Widom (2009)), possibly increasing the accessibility of the DNA by PDRM9. Moreover, independent of PDRM9, open chromatin is expected to facilitate homologous invasion, a key process for synapsis. In fact, poly-A enrichment at hotspots is also observed in species that lack this enzyme, such as plants (Horton et al, 2012; Choi et al, 2013, 2018; Wijnker et al, 2013) and yeast (Bagshaw et al, 2008), with the latter species enriched for poly-A runs ≥14 bp at hotspots (Bagshaw et al, 2008) and reviewed in (Tiemann-Boege et al, 2017). Distinguishing whether poly-A enrichment is a direct consequence of recombination and not vice versa would require further testing. However, our analysis also shows that poly-A's become increasingly rare with length, an indication that the unlimited expansion of poly-A's is constrained.

# Conclusions

How poly-A's evolve in the context of meiotic recombination is largely understudied generally, although of crucial interest also in the context of cancer research and gene regulation. Here, we provided direct experimental evidence that length asymmetry (STR heterology) and heterozygosity play a key role in the transmission and evolution of different poly-A's at hotspots. For short heterologies at poly-A's, strong meiotic conversion bias predicts their expansion. The density of longer poly-A's in hotspots, however, rapidly decays with repeat length, suggesting that an unlimited expansion is countered by some mechanism, likely alternative repair pathways (MMR versus LLR) that are activated when repeats become too long and might form longer heterologies and temporary loops. Thus, the processes that drive poly-A evolution change in nature and effect with repeat length and asymmetry.

# Materials and Methods

### Sample collection and DNA extraction

Human sperm samples were collected from anonymous donors, all of European (Austrian) descent at the IVF clinic of the MedCampus III in Linz/Upper Austria approved by the ethics commission of Upper Austria (F1–11). DNA extraction was performed with the Gentra Puregene Cell kit (QIAGEN) in 96-well plate format for identification of informative donors or in single tubes for recombinant collection. DNA extractions were from a defined number of sperm cells, ~$10^6$, as described by Arbeithuber et al (2015).

### Identification of informative donors

To find informative donors (heterozygotes), SNPs with a high heterozygosity (according to dbSNP; Sherry et al, 2001) within ~1,000-bp flanking region of the recombination hotspot were selected and allele-specific primers were designed. All our analyses were based on the chromosome assembly GRCh37/hg19. SNPs rs7201177 C/G and rs1861187 C/T upstream, and rs4786854 C/T and rs4786855 A/C downstream the hotspot (flanking SNPs) were chosen for collection of recombinant molecules (primer table in the Materials and Methods [PCR conditions for flanking SNP genotyping] section of the Supplementary Information).

Genotyping reactions were performed as described previously (Tiemann-Boege et al, 2006; Arbeithuber et al, 2015; Heissl et al, 2017) with small modifications. In short, in a total volume of 10 $\mu$l including 0.125 U OneTaq DNA Polymerase (NEB), 1× OneTaq Reaction Buffer (NEB), 0.2 mM dNTPs (Biozym Scientific GmbH), 1× SYBR Green I in DMSO (Invitrogen), 0.2 $\mu$M allele-specific primer and 0.2 $\mu$M outer primer (both Eurofins Scientific), and 10 ng total genomic DNA were used for genotyping assays. Each SNP required two separate reactions for the individual alleles. For PCR details, see the Materials and Methods (PCR conditions for flanking SNP genotyping) section of the Supplementary Information.

Informative donors were haplotyped to determine the phase of the alleles using long-range allele-specific PCR (Materials and Methods [PCR conditions for haplotyping] section of the Supplementary Information; Tiemann-Boege et al, 2006; Arbeithuber et al, 2015, 2017). In short, all 16 possible combinations of allele-specific primers were tested, with those primer pairs rendering an amplification product representing the phase of the four flanking SNPs. PCRs were set up in a volume of 10 $\mu$l containing 0.35 U of Expand Long Range Polymerase (Sigma-Aldrich), 1× Expand Long Range Standard Buffer (Sigma-Aldrich), 0.2 mM Expand Long Range dNTPack dNTPs (Sigma-Aldrich), 0.4 $\mu$M allele-specific forward primer and 0.4 $\mu$M allele-specific reverse primer (both Eurofins Scientific), 1× SYBR Green I in DMSO (Invitrogen), and 50 ng genomic DNA. Details are in the Materials and Methods (PCR conditions for haplotyping) section of the Supplementary Information.

Donors were also typed for their PRDM9 alleles (Materials and Methods [PCR conditions, purification and sequencing primers for PRDM9 variant identification] section of the Supplementary Information).

### Collection of CO and NCO events

Eight different donors were chosen for CO and NCO collection at HSII; four donors were heterozygous for the central STR rs200121160 9A/19A and four of them were homozygote for 19A. Seven donors were heterozygous for rs35094442 6A/7A, and one was homozygous for 7A (Table S1).

For the collection of single CO products, allele-specific primers for flanking SNPs were used in two rounds of nested PCR (Materials and Methods [Pooled sperm typing] section of the Supplementary Information) as previously described by Tiemann-Boege et al (2006), Arbeithuber et al (2015). The reactions contained 0.1 U Phusion HSII (Biozym Scientific GmbH), 1× HF Buffer (Biozym Scientific GmbH), 0.16 mM dNTPs (Biozym Scientific GmbH), 0.5 $\mu$M allele-specific forward primer, 0.5 $\mu$M allele-specific reverse primer (both Eurofins Scientific), and 800–1,200 molecules of genomic DNA for CO and 500 molecules for NCO collection (the Materials and Methods [PCR conditions for CO internal SNP genotyping with TaqMan] section of the Supplementary Information). The second round of PCR contained 0.1 U Phusion HSII (Biozym Scientific GmbH), 1× HF Buffer (Biozym Scientific GmbH), 0.16 mM dNTPs (Biozym Scientific GmbH), 0.5× Eva Green (Jena Bioscience), 0.5 $\mu$M allele-specific forward primer, 0.5 $\mu$M allele-specific reverse primer (Eurofins Scientific), and 2 $\mu$l of 1:10 diluted first-round PCR template for a total volume of 10 $\mu$l. The second PCR was exclusively performed for CO collection, whereas for NCOs, the first PCR product was directly used as genotyping template.

To control for amplification biases, we measured the number of "amplifiable sperm" for each donor and reciprocal recombinant described in the Materials and Methods (Testing the number of amplifiable genomes) section of the Supplementary Information. In short, one to two non-recombinant molecules per reaction were amplified in *Escherichia coli* carrier DNA using the same PCR conditions as for the recombinant collection. We then estimated the amplifiable sperm by correcting the sperm DNA concentration measured spectrophotometrically by the number of effective positive reactions. The correction factors across experiments were within 0.10–0.35 (Table S11) and differed only slightly between donors and experiments. The total number of meiosis used per reaction was corrected by the effectively amplifiable number of meiosis for each donor.

### Genotyping of CO and NCO events

COs were genotyped with TaqMan PCR for the SNPs rs12102448 A/G, rs112051149 C/T, rs72778219 C/T, and rs8060928 C/T. A total of 10 $\mu$l were used containing 2 $\mu$l of 1:1,000 diluted second PCR product, 0.15 U peqGold Hot Taq Polymerase (PEQLAB), 3 mM MgCl$_2$, 0.2 mM dNTPs, 0.4 $\mu$M forward primer, 0.4 $\mu$M reverse primer, 0.2 $\mu$M FAM-labelled probe (allele 1), and 0.2 $\mu$M HEX-labelled probe (allele 2; all Eurofins Scientific; Materials and Methods [PCR conditions for CO internal SNP genotyping with TaqMan] section of the Supplementary Information). SNP rs12102452 A/C, rs35094442 6A/7A, and the rs200121160 9A/19A microsatellite were genotyped with allele-specific primers with 0.06 U Phusion HSII polymerase (Biozym Scientific GmbH), 1× HF buffer (Biozym Scientific GmbH), 0.2 $\mu$M dNTPs (Biozym Scientific GmbH), 1× SYBR Green I in DMSO (Invitrogen), 0.4 $\mu$M open forward primer, 0.4 $\mu$M allele-specific reverse primer, and 2 $\mu$l of 1:1,000 diluted second PCR product (Materials and Methods [PCR conditions for CO and NCO internal SNP genotyping with allele-specific primers] section of the Supplementary Information).

NCOs were genotyped with iTaq DNA polymerase (Bio-Rad) and Phusion HSII (Biozym Scientific GmbH) for the 9A/19A microsatellite.

For the iTaq protocol, 10 $\mu$l final volume containing 0.25 U iTaq Polymerase (Bio-Rad), 1× reaction buffer (Bio-Rad), 1.5 mM MgCl$_2$ (Bio-Rad), 1× SYBR Green I in DMSO (Invitrogen), 0.4 $\mu$M allele-specific forward primer, 0.4 $\mu$M allele-specific reverse primer, and 2 $\mu$l of 1:10 diluted PCR product were used. The genotyping of 9A/19A microsatellite required 10 $\mu$l total volume containing 0.06 U Phusion HSII polymerase (Biozym Scientific GmbH), 1× HF buffer (Biozym Scientific GmbH), 0.2 $\mu$M dNTPs (Biozym Scientific GmbH), 1× SYBR Green I in DMSO (Invitrogen), 0.4 $\mu$M open forward primer, 0.4 $\mu$M allele-specific reverse primer, and 2 $\mu$l of 1:100 diluted PCR product (Materials and Methods [PCR conditions for CO and NCO internal SNP genotyping with allele-specific primers] section of the Supplementary Information and Heissl et al, 2017). Data analysis was performed as described in the Materials and Methods (Data analysis) section of the Supplementary Information.

### Data analysis

#### *Test for biased gene conversion*
Statistical analysis of the CO and NCO data was performed using R v. 3.3 (https://www.r-project.org/), with Poisson tests using the exactci package v 1.3-3 and the Cochrane–Mantel–Haensel test with metafor 2.1-0. CO and NCO rates were checked for potential donor heterogeneity with a generalized linear model with a quasibinomial error model [Full model: (Number of CO/NCO molecules, other molecules) ~ donor identity + donor type (Ht/Ho) + type (NCO/CO)]. As no significant effect of donor was found, we instead used a simple Poisson test for the analysis.

#### *Permanganate/S1 footprinting data analysis for identifying ssDNA regions*
Illumina sequencing data were extracted from National Center for Biotechnology Information Sequence Read Archive (ID: SRA072844) for human Burkitt's lymphoma line Raji performed by Kouzine et al (2017). In short, oxidation of unpaired thymidines in ssDNA regions were screened by adding permanganate to living cells and, thus, stabilizing the single-stranded state and increasing the sensitivity of these regions for digestion by ssDNA nucleases (S1 nuclease). The digested regions were then sequenced with Illumina single-ended reads (50-bp long). The reads were mapped with BWA-MEM (Li & Durbin, 2010) to the human reference genome GRCh37/hg19.

#### *Genome-wide poly-A analysis*
We investigated the distribution of poly-A's over the whole genome using the full genome sequence for *Homo sapiens* as provided by University of California, Santa Cruz (UCSC; GRCh37/hg19) from the R-package *BSgenome.Hsapiens.UCSC.hg19* (Bioconductor Dev Team, 2014). For this purpose, we searched every human chromosome for poly-A's of length six or longer and matched this information with the DSB map genome coordinates provided by (Pratto et al, 2014). We then classify the poly-A's with respect to their location within the DSB into different *regions*, where poly-A's can either be in a "hotspot" (referring to the DSB-coordinates of Pratto et al, 2014 with additional 500 bp left and right), in "hotspot-flanking segments" or "outside hotspots." The hotspot-flanking segments were split into five adjacent sliding windows, each 1 kb left and right of the hotspot region. The first segment starts at the left and right

DSB hotspot thresholds (±500 bp) and spans 1 kb left and right of these two boundaries. We construct all further segments analogously using the new limits of adjacent regions sequentially up to the fifth flanking region. The analysis corrects each region by the corresponding total segment lengths (number of nucleotides).

### SGDP analysis

The lobSTR reference sites called in from the SGDP were downloaded from http://strcat.teamerlich.org/download (Willems et al, 2014; Mallick et al, 2016). We restricted our analysis to genotypes of the 300 individuals of West Eurasian descent (likely carriers of PRDM9 allele A) and filtered out low-quality calls as described in (Mallick et al, 2016). Because the default minimal tract length of the called STRs was set to 11 nt, no data for poly-A's between 6 and 10 A's were available. Genome coordinates of perfect poly-A repeats were extracted, whereas imperfect poly-A's were removed. Repeats with more than 26 A's were set to a length of 26.

The heterozygosity for each site was calculated as $1 - \sum_{i=1}^{n} P_i^2$, where $P_i$ is the frequency of the $i$th allele and $n$ is the number of different alleles at the locus. Further variables that we analyzed for each locus were the difference in length between the longest and shortest allele (allelic asymmetry), length differences of alleles (steps in length), and the number of different alleles.

## Supplementary Information

## Acknowledgements

Open access funding was provided by the Austrian Science Fund (FWF). This work was also supported by the "Austrian Science Fund" (FWF) P27698-B22 to I Tiemann-Boege. A Heissl was funded by a DOC Fellowship (24529) of the Austrian Academy of Sciences at the Institute of Biophysics, Johannes Kepler University. B Arbeithuber was funded by a DOC Fellowship (23722) of the Austrian Academy of Sciences at the Institute of Biophysics, Johannes Kepler University. We are grateful to Simon Myers and Nick Altemose for the correspondence about PRDM9 targets.

### Author Contributions

A Heissl: conceptualization, investigation, methodology, and writing—original draft, review, and editing.
AJ Betancourt: formal analysis, investigation, methodology, and writing—original draft, review, and editing.
P Hermann: formal analysis, investigation, methodology, and writing—original draft, review, and editing.
G Povysil: formal analysis, investigation, and writing—original draft.
B Arbeithuber: formal analysis, investigation, and methodology.
A Futschik: conceptualization and funding acquisition.
T Ebner: resources.
I Tiemann-Boege: conceptualization, supervision, funding acquisition, investigation, project administration, and writing—original draft, review, and editing.

### Conflict of Interest Statement

The authors declare that they have no conflict of interest.

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
