## [Reviewer comments · Life Science Alliance]

Life Science Alliance

The impact of poly-A microsatellite heterologies in meiotic recombination

Angelika Heissl, Andrea Betancourt, Philipp Hermann, Gundula Povysil, Barbara Arbeithuber, Andreas Futschik, Thomas Ebner, and Irene Tiemann-Boege

DOI: [10.26508/lsa.201900364](https://doi.org/10.26508/lsa.201900364)

Corresponding author(s): Irene Tiemann-Boege, Johannes Kepler University, Linz, Austria and Angelika Heissl, Institute of Biophysics, Johannes Kepler University Linz

Review Timeline:	Submission Date:	2019-03-04
	Editorial Decision:	2019-03-20
	Revision Received:	2019-03-27
	Accepted:	2019-03-29

Scientific Editor: Andrea Leibfried

Transaction Report:

Please note that the manuscript was previously reviewed at another journal and the reports were taken into account in the decision-making process at Life Science Alliance. Since the original reviews are not subject to Life Science Alliance's transparent review process policy, the reports and author response cannot be published.

March 20, 2019

RE: Life Science Alliance Manuscript #LSA-2019-00364-T

Irene Tiemann-Boege
Institute of Biophysics
Johannes Kepler University
Linz, Austria

Dear Dr. Tiemann-Boege,

Thank you for submitting your revised manuscript entitled "Length asymmetry of poly-A microsatellites strongly influences meiotic recombination hotspots" to Life Science Alliance. Your work was previously reviewed at another journal and the editors transferred those reports to us with your permission.

The reviewers who assessed your work elsewhere before noted a few technical issues and thought that the overall advance provided remains somewhat limited. The latter is not a concern for publication in Life Science Alliance, and as you can see below, original reviewer #2 appreciates the changes introduced to address the technical issues. We would thus be happy to publish your paper in Life Science Alliance pending final revisions necessary to meet our formatting guidelines:

- please include a summary blurb in our submission system
- please add the following callouts to the manuscript text: Figure 2A, 3C, 3D, Table S4
- please upload your manuscript text as a docx file that includes all figure legends (also S Figure legends), table legends as well as all tables
- please upload all figure files without legends and as individual files, also the supplementary figures
- as this is a primary research article, I would like to suggest to change the title to: The impact of poly-A microsatellite heterologies in meiotic recombination

A. FINAL FILES:

B. MANUSCRIPT ORGANIZATION AND FORMATTING:

Sincerely,

Andrea Leibfried, PhD
Executive Editor
Life Science Alliance
Meyerhofstr. 1
69117 Heidelberg, Germany

t +49 6221 8891 502
e a.leibfried@life-science-alliance.org
www.life-science-alliance.org

Reviewer #2 (Comments to the Authors (Required)):

My comments on a previous manuscript version were answered appropriately.

March 29, 2019

RE: Life Science Alliance Manuscript #LSA-2019-00364-TR

Irene Tiemann-Boege
Johannes Kepler University, Linz, Austria
Institute of Biophysics
Gruberstrasse 40
Linz 4020
Austria

Dear Dr. Tiemann-Boege,

Thank you for submitting your Research Article entitled "The impact of poly-A microsatellite heterologies in meiotic recombination". It is a pleasure to let you know that your manuscript is now accepted for publication in Life Science Alliance. Congratulations on this interesting work.

DISTRIBUTION OF MATERIALS:

Again, congratulations on a very nice paper. I hope you found the review process to be constructive and are pleased with how the manuscript was handled editorially. We look forward to future exciting submissions from your lab.

Sincerely,
